# Analytical Quality by Design (AQbD) Approach to the Development of Analytical Procedures for Medicinal Plants

**DOI:** 10.3390/plants11212960

**Published:** 2022-11-02

**Authors:** Geonha Park, Min Kyoung Kim, Seung Hyeon Go, Minsik Choi, Young Pyo Jang

**Affiliations:** 1Department of Life and Nanopharmaceutical Sciences, Graduate School, Kyung Hee University, Seoul 02447, Korea; 2Division of Pharmacognosy, College of Pharmacy, Kyung Hee University, Seoul 02447, Korea; 3Department of Biomedical and Pharmaceutical Sciences, Graduate School, Kyung Hee University, Seoul 02447, Korea; 4Department of Integrated Drug Development and Natural Products, Graduate School, Kyung Hee University, Seoul 02447, Korea

**Keywords:** analytical quality by design (AQbD), medicinal plants, natural products, analytical tools, lifecycle management, method operable design region (MODR)

## Abstract

Scientific regulatory systems with suitable analytical methods for monitoring quality, safety, and efficacy are essential in medicinal plant drug discovery. There have been only few attempts to adopt the analytical quality by design (AQbD) strategy in medicinal plants analysis over the last few years. AQbD is a holistic method and development approach that understands analytical procedure, from risk assessment to lifecycle management. The enhanced AQbD approach reduces the time and effort necessary to develop reliable analytical methods, leads to flexible change control through the method operable design region (MODR), and lowers the out-of-specification (OOS) results. However, it is difficult to follow all the AQbD workflow steps in the field of medicinal plants analysis, such as defining the analytical target profiles (ATPs), identifying critical analytical procedure parameters (CAPPs), among others, because the complexity of chemical and biological properties in medicinal plants acts as a barrier. In this review, various applications of AQbD to medicinal plant analytical procedures are discussed. Unlike the analysis of a single compound, medicinal plant analysis is characterized by analyzing multiple components contained in biological materials, so it will be summarized by focusing on the following points: Analytical methods showing correlations within analysis parameters for the specific medicinal plant analysis, plant raw material diversity, one or more analysis targets defined for multiple phytochemicals, key analysis attributes, and analysis control strategies. In addition, the opportunities available through the use of design-based quality management techniques and the challenges that coexist are also discussed.

## 1. Introduction

As a major source of natural products, medicinal plants have been used for thousands of years as important candidates for herbal medicinal use due to their diverse sources of therapeutic phytochemicals [1]. The US Food and Drug Administration (USFDA) botanical drugs, European herbal products (HMP), Chinese herbal medicines (TCM), and other botanical drugs in other countries are drugs that generally contain natural products. The intended therapeutic efficacy of these drugs is likely to be realized through a multi-component, multi-targeted mode of action with a heterogeneous mixture [2]. The complexity of natural products from medicinal plants has been expected to deliver the synergistic potential of phytochemicals as active pharmaceutical ingredients [3]. As herbal ingredients are sourced from nature (wild and/or farm-grown), they are exposed to a variety of environmental influences. Therefore, considerable attention must be paid to quality-changing factors, such as pesticide residues, heavy metal contamination, use of different plant parts, geographic influences, post-harvest treatment, among others. [4]. In addition to these various factors, the quality of pharmaceuticals developed from medicinal plants is also affected by the production process, such as the choice of extraction process. Various extraction methods and procedures can be established according to the desired chemical substances and biomarkers in the herbal raw material. Optimized extraction methods generate sufficient amounts of bioactive chemicals in the extract, which is the basis for the expected biological properties [5]. In general, chemical markers are used to collectively monitor the constancy of quality, sufficient efficacy, and safety throughout the entire process of oriental medicine development [6].

The pharmaceutical industry relies on analytical data for multiple parts of the pharmaceutical process, from raw materials to final pharmaceuticals [7]. Since all production processes affect quality consistency, it is very important to have a monitoring and control system for each process. Validated analytical methods are considered to be effective tools for the systematic approach of tracking target chemicals, quality control protocols in drug development and manufacturing [8,9], and risk-based life cycle management [10]. The International Council for Harmonization of Technical Requirements for Pharmaceuticals for Human Use (ICH) defines risk as the combination of the probability, severity, and detectability of the harm to the quality of the active pharmaceutical ingredient (API) and medicinal product, thus announcing the ICH guideline Q9 on quality risk management (QRM) [11]. The QRM for an analytical procedure provides systematic access to deep understanding, and to check for risks, monitor, and control the quality of the data that is acquired in each method performance across the entire life cycle [12]. Additionally, the United States Pharmacopeia (USP) published the General Chapter <1220> “Analytical Procedure Life Cycle”, which is a framework for the overall evidence based approach to establish an analytical procedure, including continuous modification and improvement through circulation in the life cycle [13]. The analytical method established by applying life cycle management has regulatory flexibility [14]. Unlike the conventional analytical procedures (e.g., quality-by-testing (QbT) methodology, trial-and-error approach, or one-factor-at-time (OFAT) investigation) [15], which would be fixed to a consistent method with associated performing criteria, the enhanced approach could facilitate changes in analytical parameters within proven robust design space to support continuous improvement, even without the requirement of the revalidation protocol [14,16,17]. The holistic understanding and control strategy to the analytical procedure can be practically achieved through an analytical quality by design (AQbD) concept, which is described in Step 2 draft of ICHQ14 on the analytical procedure development [18]. In ICHQ14, it is also worth emphasizing that AQbD is called the enhanced approach.

In the past 20 years since the FDA and ICH leadership, Quality by Design (QbD) principles have been increasingly introduced across the pharmaceutical industry [19,20]. Notably, the definition of QbD is defined by the ICH guideline Q8 (R2) as “A systematic approach to development that begins with predefined objectives and emphasizes product and process understanding and process control, based on sound science and quality risk management” [21]. Increasing testing or testing at different stages does not necessarily improve product quality, robust manufacturing processes, or efficient development. However, quality can be obtained more effectively through a QbD systematic process that is scientifically designed to achieve its objectives and includes an in-depth understanding of regulatory properties and pharmaceutical quality system (PQS) [22]. AQbD is an application of the QbD concept to the process of developing an analytical method, which plays a critical role in monitoring the quality of every step in the medicinal production system [23]. AQbD begins with the in-depth analytical method by understanding and building specific objectives through sound–science and quality risk management. For the specific objectives, researchers predefined the analytical target profile (ATP), which is a “a prospective description of the desired performance of an analytical procedure that is used to measure a quality attribute, and it defines the required quality of the reportable value produced by the procedure” [24]. Herein, specific indicators and/or characteristics of the method performance, which have a significant influence on the specific analytical procedure, defined as critical analytical procedure attributes (CAPAs), are established. While analytical parameters that have an intimate effect on CAPAs, such as environmental conditions, sample characteristics, technical parameters, measurement, and instrument configuration, are determined through a risk assessment approach [24]. In general, there are three risk evaluation tests: Traffic light risk analysis, Ishikawa diagram, and failure modes and effects analysis, which are confirming the cause–effect relationship between input parameters and analytical quality CAPAs [8]. Parameters assessed to have the highest risk are selected as critical analytical procedure parameters (CAPPs) and should be controlled or monitored to confirm that the analytical protocol meets the desired quality [25]. In order to establish the initial analytical control strategy, the design of experiments (DoE) approach can be utilized, which is a mathematical approach that uses statistical tools [26]. However, because of the large number of parameters involved, there is a high risk that analytical performance may become critical, especially if the CAPPs affect the quantitative results of the target substances. When the effects of the parameters affecting the selected CAPAs and the effect within those parameters are complex, the procedure of simplifying CAPPs through the DoE strategy is introduced first and then the screening design is used to optimize the analytical process [27]. In particular, the results of the screening design show the implementation of the dynamic factor-to-response relationship of CAPPs, through the three-dimensional (3D) space of the response surface design and expresses it as a polynomial equation [28]. This process is called an optimization step, and the method operable design region (MODR) is derived, rather than a fixed analytical method. Technical flexibility can be achieved through the MODR, which is a scientific warrant that it can be sufficiently tolerated without the further approval of procedures when the procedure changes within the fulfilled analytical performance criteria of the working region [14,28]. After this, a working point is selected within the MODR to perform the procedure validation for quality control methods based on various criteria (e.g., specificity and sensitivity for identification, accuracy and precision for quantification). Finally, a planned set of control strategies should be discussed and should encompass the whole AQbD procedure, which ensures continuous satisfaction of the required ATP criteria by life cycle management [29]. This step includes both the routine monitoring of the achieved data from performing in every batch and the performance evaluation after changes to judge whether the purpose of the analysis procedure is fit [30]. The AQbD workflow is shown in Figure 1, representing the analytical quality control method life cycle.

As shown in Figure 2, the attempts to apply the AQbD concept in pharmaceutical science have increased gradually over the past decade. The number of scientific publications per year related to AQbD has increased rapidly between 2012 and 2021 in the Scopus database. Interestingly, there were very few AQbD papers published at this time on medicinal plants. This trend seems to be due to the complexity and diversity of the secondary metabolites contained in medicinal plants. Moreover, it may be partly because it is difficult to implement each step of the AQbD workflow using the multi-components multi-targets principle, which is considered to be the mechanism of action of medicinal plants. 

This review aims to present the currents, prospects, and challenges of the AQbD methodology applied to the field of analytical method development in medicinal plants. It consists of a brief overview of the selected analytical techniques, target plant raw materials, defined API from multi-phytochemicals, key analytical attributes, and correlation within analytical parameters to optimize the analytical control strategy, as well as analytical procedure life cycle trials. While the application of AQbD has been evolving in the development of medicinal plant analysis procedures, we reviewed the practice cases published in the article from 2012 to focus on what achievements were made in carrying out the essential steps of AQbD and what acted as barriers. The terminology and abbreviations in this paper are sorted by each subtitle and are listed in Table 1.

## 2. *Fit-for-Purpose*: Analytical Target Profiles (ATPs)

The first step in AQbD (Figure 1) is to define the ATP for the intended analytical procedure. ATP is a summary of the well-defined prospective requirements; an analytical procedure must meet the required quality of the measurement thought accurate assessment [31]. Since it does not intend to be a method and/or analytical technique, it provides the flexibility to apply any analytical procedure if it meets the ATP criteria [31,32]. The ATP includes all the elements, such as target sample, target APIs, sample preparation, required analytical technique, instrument requirement, method requirement, target application, reportable quality attributes, and critical analytical characteristics. The target application includes general assays, such as identification, separation, and quantification of the potency test for a drug substance and other specification tests, like impurity profile and residual solvents, to ensure the efficacy and safety of the final drug product [33]. In particular, when analyzing herbal plants, the goal may be species discrimination using chromatographic fingerprints or correlation analysis between the batches in order to track the quality change according to the cultivation environment, collection period, storage, etc.

Several reports published regarding the application of AQbD to the development of analytical methods for plant resources from 2016 to 2021 are summarized with each characteristic in Table 2. In this review, the ATP of each paper is classified into three categories: APIs, analytical techniques, and method requirements. Analytical procedures were developed for the purpose of various classes of APIs in various medicinal plants. Most of the papers were aimed at the separation and quantification of the APIs by each analytical technique, therefore liquid chromatography, which is popularly used for the separation of natural substances, was mainly utilized. This technique focused on obtaining chromatograms rather than obtaining the spectra of APIs to analyze the desired APIs in the oceans of various phytochemicals. Classes of flavonoids [34,35], terpenoids [36], quinones [37], saponins [38,39], and phenolic compounds, such as curcuminoids [40] and coumarins [36] were analyzed by (Ultra) High Performance Liquid Chromatography-Ultraviolet/Visible (photodiode array detector) ((U)HPLC-UV/Vis (PDA) techniques. Mass spectrometry was employed for the analysis of terpene lactones [41] and polyphenols [42], and supercritical fluid chromatography was utilized for the analysis of cannabinoids [43]. Various detection methods, such as the refractory index (RI) [44], evaporative light scattering detector (ELSD) [45] and sample preparation processes [46,47], were applied for the analysis of sugar and its derivatives because they are not easily detected by the UV/Vis spectrometer.

## 3. *Backbone-of-AQbD*: Risk Assessment

After the ATP is established, analytical procedure attributes (APAs) that affect analytical performance quality should be selected. These APAs are key variables tracked by quantifiable analytical outputs and must be within reasonable bounds to ensure the expected quality of the analytical process [35]. In general, CAPAs are determined through risk assessment based on prior knowledge and a scientific process [12]. Risk assessment aims to identify and rank analytical procedure parameters (APPs) that affect method performance, such as analytical procedure attributes (APAs) and ATP compliance [12,24]. Therefore, it forms the first impression and the backbone of AQbD to connect ATP, APAs, APPs, and the control strategy [24]. After implementing the risk assessment, a prioritized list of “critical” analytical procedure attributes (CAPAs) and “critical” analytical procedure parameters (CAPPs) is obtained [25]. The principal tools used to perform the risk assessment are flowcharts, check sheets, process mapping, Ishikawa diagram (as known as fishbone diagram or cause and effect diagram), failure mode effects analysis (FMEA), and many others [11]. After this, modelling and statistical evaluation of the analytical procedure with CAPPs provides the method operable design region (MODR), which can be the base of the regulatory flexibility of the permission to change the analytical procedure within the designed region. In the AQbD approach for medicinal plants, the Ishikawa diagram have been dominantly utilized to determine the latent CPPs, followed by FMEA and the risk estimation matrix (known as the traffic light risk analysis) (Table 2) [9,12]. 

Risk assessment is the key step to the success of AQbD and is a precursor to proper design space studies. It is also critical to show the focus on procedure development efforts and their effectiveness. AQbD sometimes fails due to the inappropriate risk assessment, even before the design space stage is reached. Therefore, the risk assessment should be updated at each step from the initial stage of development to continuous monitoring in the analytical procedure lifecycle.

### 3.1. Ishikawa Diagram

The Ishikawa diagram, also called the fish-bone diagram or cause and effect diagram, is a visualization tool used to categorize the possible causes of a problem [21]. It can dissect the risk in various categories related, for example, to sample preparation, instrumentation, materials, personnel factors, and environmental conditions in order to determine the root causes of the problem. For the most of chromatographic methods, risk assessment is performed using the Ishikawa diagram with 6M branches (Mother nature, Machine, Materials, Method, Measurement, and Man) [34,37,41,42,43,44,46]. This method allows us to visualize and simultaneously show various parameters that have the potential to influence the analysis results, however it is not able to determine the association between parameters.

### 3.2. Failure Modes Effects Analysis (FMEA)

FMEA is a proactive qualitative and systemic risk analysis that identifies and ranks all potential problems (failure modes) and their repercussions (effects analysis) [50]. The purpose of the FMEA is to take steps to eliminate or reduce defects, starting with the highest-priority defects. Failure modes are prioritized according to how serious their consequences are (S), how frequently they occur (O), and how easily they can be detected (D). The risk priority number (RPN) is calculated by multiplying the scores of S, O, and D, each between 1 and 10, with higher numbers indicating higher risk [51,52]. Kim et al. [34] and Zhang et al. [35] used FMEA as a successor to the fishbone diagram to select high-risk factors. In both studies, HPLC-PDA was used to analyze many phytochemicals at the same time, and since various parameters interact in a complex way, FMEA was used for the purpose of densely and precisely identifying higher risk parameters through the scoring process. As the various latent CAPPs’ risk levels are expressed numerically, priorities of factors can be easily compared.

### 3.3. Risk Estimation Matrix

Risk estimation matrix (or traffic light or heat map) helps to visualize the level of risks using color codes [27]. Red means major or catastrophic risks, yellow means moderate risks, and green indicates minor or insignificant risks. It has the advantage of being able to grasp the effect of each parameter on each CAPAs at a glance. Tiwari et al. [37] and Kim et al. [36] used the risk estimation matrix to estimate the correlation between CAPAs and latent CAPPs in the HPLC analysis. When CAPPs are determined without risk assessment based on prior knowledge and prior studies, a few preliminary experiments need to be conducted to test the effectiveness of CAPPs [47].

## 4. *Sort-the Main Effects-Out*: Design of Experiments (DoE) for Screening Factors

CAPAs and CAPPs can be derived directly from risk assessment, but if there are too many attributes and parameters to consider, a Design of Experiment (DoE) concept can be applied to select the significant few among a number of potential CAPAs and CAPPs. It is almost impossible to understand and control all the effects of variables that affect the response when conducting experiments with traditional methods, such as QbT methodology, OFAT investigation, and trial-and-error approach [53,54]. The purpose of DoE is to establish an experiment plan to obtain ‘maximum information’ with ‘minimum experiment’ through statistical methods [55]. It is a systematic process in which a researcher designs an experiment or series of experiments, collects data, and derives results in the form of graphs and mathematical equations [56]. Using these equations, the outcomes of the experiment can be predicted and optimized [57]. The application of the DoE concept is economical and efficient because it leads to good experimental results with a relatively limited number of experiments for various experimental conditions [27]. The DoE for factor screening (called screening DoE forward) helps to identify the factors that influence responses and evaluate all possible factors that may affect the analytical procedure attributes to understand the qualitative, quantitative, and comprehensive aspects of the factors [58]. In several research studies, it was shown that full factorial design (FFD), fractional factorial design (fFD), and Placket-Burman design (PBD) are mainly used for screening DoE [59,60,61,62,63,64,65]. The characteristics of screening DoE models are summarized in Table 3.

### 4.1. Full Factorial Design (FFD)

Full factorial design (FFD) is used to understand the main effects of the factors, as well as their interactions, without being confused, usually in two to five factors discussed [14,66]. In the screening design process, two-level FFD is the design first considered because of its economic efficiency [54]. The factors are represented by capital letters and a high level of factors is expressed as (+) or +1 and a low level of factors is expressed as (−) or −1 and a middle level of factors is expressed as 0 [67]. The two-level FFD matrix for two, three, and four experimental factors are represented in Table 4. All possible combinations of factors are contained in the model and the equation of the number of experiments is expressed as level*^factors^* (Figure 3a,b) [55]. Among the papers that applied AQbD to natural product analysis, Kim et al. [34] utilized four-factors and two-level (2^4^) FFD to identify which variables mainly affect the number of flavonoid peaks of Genkwa Flos out of column temperature, flow rate, injection volume, and gradient slope of mobile solvents.

### 4.2. Fractional Factorial Design (fFD)

It is sometimes not efficient to assess all the effects and interactions among the experimental factors when the number of factors to be evaluated increases [68], for example, 128 experimental runs are required to identify seven factors under FFD and 1024 runs to identify 10 factors [55]. The effect of individual factors and the interaction between variables affect the experimental response, and even if it is statistically significant, knowledge and understanding of these factors may be insufficient [68]. In this case, it is desirable to obtain information on the factors that mainly affect the experimental response and perform a minimal experiment at the expense of less meaningful higher-order interactions [68]. Fractional factorial design (fFD) is one of the commonly used screening DoE models when the number of experimental factors is four or more, due to its ability to evaluate the effects of many experimental factors with a relatively small number of experiments [66]. The fFD is designed by dividing the FFD by 2^n^, and Figure 3c and d illustrate the three-dimensional model of fFD with three experimental factors and its complementary model [67]. Since fFD does not require the experimental design of all points of the cubic matrix, it has the advantage of reducing time, but has the disadvantage that the main effects and interactions of the variables can be confounded [67]. In order to avoid this confounding, the number of design generator, a concept for understanding the confounding effects used to separate the full factorial into fractional factorial, should be considered carefully to select an appropriate fFD [55,68]. The number of experiments in fFD is 2^k−p^, where k is the number of experimental factors, and p is the number of design generators used to separate the design [68]. For example, when designing an experiment for four experimental factors, a half-fraction factorial design (2^4−1^ = 8 experimental runs) may be chosen and when designing an experiment for five experimental factors, quarter-fraction factorial design (2^5−2^ = 8 experimental runs) may be chosen [54]. The fFD matrix for three, four, and five experimental factors are represented in Table 5. Shao et al. [45] applied two-level fFD to screen seven parameters and found three parameters as CAPPs when analyzing the sugar contents of *Codonopsis* Radix extract and *Astragali* Radix extract using HPLC-ELSD.

### 4.3. Plackett-Burman Design (PBD)

Plackett-Burman design (PBD) is a two-level, mathematically-derived multiple of four screening design, introduced by R.L. Plackett and J.P. Burman in 1946 [69]. PBD is used to study *N* − 1 factors in *N* experiments, proposing experimental designs with more than seven factors, especially multiples of four (at least 8), i.e., 8, 12, 16, 20, etc., and is suitable for studying factors up to 7, 11, 15, 19, etc. [70]. A two-level PBD is similar to a saturated fFD because designing the seven factors with 2-level PBD and designing with a saturated fFD (2^7−4^) are similar [58,70]. When designing an experiment with PBD for eleven experimental factors, twelve experimental runs are designed (Table 6) [58,70]. PBD is useful for screening, where unbiased estimates of all the main effects can be provided through as few experiments as possible, but it is difficult to evaluate the interaction between the main effects [16,21]. Hence, PBD is widely used for screening variables of medicinal plants analysis [35,41,47,49]. Zhang et al. sed PBDs to select four CAPPs under fifteen runs designed from seven factors with repeating three center points (CP) [41]. In another PBD study, four CAPPs were selected under twelve runs designed from eight factors without testing CP [35]. When the number of factors that can, potentially, be examined in PBD (*N* − 1) exceeds the number of factors to be examined, the remaining columns are defined as dummy factor columns, which is a hypothetical variable that varies between levels −1 and +1 with no physicochemical meaning [62]. Sheng-Yun and co-researchers utilized a PBD to select three CAPPs among five latent APPs aiming at maximizing critical resolution, minimizing analysis time, and minimizing peak width of the major alkaloids of *Coptis chinensis* [49]. Wang et al. [47] applied PBDs to screen the main effects of both solid phase extraction (SPE) and HPLC-UV/ELSD analysis process for the quantification of nine bioactive compounds in Shenqi Fuzheng injection consisting of *Codonopsis pilosula* and *Astragalus membranaceus*.

**Table 6 plants-11-02960-t006:** The matrix of Plackett-Burman design (PBD) for eleven experimental factors.

No.	X_1_	X_2_	X_3_	X_4_	X_5_	X_6_	X_7_	X_8_	X_9_	X_10_	X_11_
1	+1	−1	+1	−1	−1	−1	+1	+1	+1	−1	+1
2	+1	+1	−1	+1	−1	−1	−1	+1	+1	+1	−1
3	−1	+1	+1	−1	+1	−1	−1	−1	+1	+1	+1
4	+1	−1	+1	+1	−1	+1	−1	−1	−1	+1	+1
5	+1	+1	−1	+1	+1	−1	+1	−1	−1	−1	+1
6	+1	+1	+1	−1	+1	+1	−1	+1	−1	−1	−1
7	−1	+1	+1	+1	−1	+1	+1	−1	+1	−1	−1
8	−1	−1	+1	+1	+1	−1	+1	+1	−1	+1	−1
9	−1	−1	−1	+1	+1	+1	−1	+1	+1	−1	+1
10	+1	−1	−1	−1	+1	+1	+1	−1	+1	+1	−1
11	−1	+1	−1	−1	−1	+1	+1	+1	−1	+1	+1
12	−1	−1	−1	−1	−1	−1	−1	−1	−1	−1	−1

## 5. *Make-the Best Conditions*: Design of Experiments (DoE) for Optimization

The CAPPs selected through the screening DoE process can be studied further and optimized through the optimization DoE process. DoE for CAPPs optimization (called Optimization DoE forward) is divided into symmetric and asymmetric models [71]. The symmetrical design includes a symmetrical experimental area and the CP is measured three to five times to evaluate the experimental errors [53,62]. Full factorial design (FFD) [72], central composite design (CCD) [53], Box-Behnken design (BBD) [73], Taguchi design [74], and Doehlert design [62] are considered as the symmetrical designs [58]. The asymmetrical designs, such as D-optimal design, are applied when it is necessary to evaluate an asymmetrical experimental area [53,62]. To optimize the analytical procedure for natural products, 3^3^-level FFD [40,43,44,46], CCD [34,35,36,39], BBD [37,41,45,47,49], and Doehlert design [42,48] are mainly conducted. The characteristics of the optimization DoE models are summarized in Table 3, and four commonly used optimization models are described in detail.

### 5.1. Full Factorial Design (FFD)

FFD can not only be applied for screening DoE, but also for optimization DoE, and when FFD is used for optimization DoE, three-level FFD is mainly considered because it allows for modeling of the complex response surface [54] and testing all the level (−1, 0, and +1) combinations of all experimental factors [62]. FFD for optimization is usually used when the number of factors is few, such as two or three, because all combinations of factors must be tested and so the number of experiments needed increases dramatically as the number of factors increases [66,67,73]. The number of experiments in 3-level FFD is 3*^factors^* (level*^factors^*), i.e., the experiment with two factors requires nine runs (3^2^) and the experiment with three factors requires 27 runs (3^3^) [62]. The matrix of 3-level FFD for three factors is summarized in Table 7 and Figure 4a. Silva et al. [44] designed 3^3^-FFD to evaluate three CAPAs (area, resolution, and asymmetry of the objective peaks) for the separation, identification, and quantification of sugar from sugarcane honey using HPLC-RI and three CAPPs (flow rate, column temperature, and ratio of mobile solvent) were selected from the risk assessment. Parab Gaonkar et al. [40] applied 2^2^-FFD without testing CP to optimize concentration of orthophosphoric acid in the mobile phase and the ratio of mobile phases for the quantification of curcuminoids in *Curcuma longa* using HPLC-UV. Unusually, FFD can be performed by excluding CP because CP cannot be set due to the factors’ characteristics [43,46]. Deidda et al. [43] used 2^3^-FFD to analyze nine cannabinoids from *Cannabis* species and Silva et al. [46] utilized three different FFD models, which are 1-level with 2-factors (1^2^), 2-level with 3-factors (2^3^), and 1-level with 4-factors (1^4^) for the separation and quantification of sugars from *Saccharum officinarum* by microextraction by packed sorbent (MEPS)-UHPLC-PDA analysis.

**Table 7 plants-11-02960-t007:** The matrix of 3^3^-Full Factorial Design (3^3^-FFD), Central Composite Design (CCD), and Box-Behnken Design (BBD).

3^3^−Full Factorial Design				Central Composite Design				Box−Behnken Design			
No.	X_1_	X_2_	X_3_	No.	X_1_	X_2_	X_3_	No.	X_1_	X_2_	X_3_
1	−1	−1	−1	1	−1	−1	−1	1	−1	−1	0
2	−1	−1	0	2	+1	−1	−1	2	+1	−1	0
3	−1	−1	+1	3	−1	+1	−1	3	−1	+1	0
4	−1	0	−1	4	+1	+1	−1	4	+1	+1	0
5	−1	0	0	5	−1	−1	+1	5	−1	0	−1
6	−1	0	+1	6	+1	−1	+1	6	+1	0	−1
7	−1	+1	−1	7	−1	+1	+1	7	−1	0	+1
8	−1	+1	0	8	+1	+1	+1	8	+1	0	+1
9	−1	+1	+1	9	−1.68	0	0	9	0	−1	−1
10	0	−1	−1	10	+1.68	0	0	10	0	+1	−1
11	0	−1	0	11	0	−1.68	0	11	0	−1	+1
12	0	−1	+1	12	0	+1.68	0	12	0	+1	+1
13	0	0	−1	13	0	0	−1.68	13	0	0	0
14	0	0	0	14	0	0	+1.68	14	0	0	0
15	0	0	+1	15	0	0	0	15	0	0	0
16	0	+1	−1	16	0	0	0				
17	0	+1	0	17	0	0	0				
18	0	+1	+1	18	0	0	0				
19	+1	−1	−1	19	0	0	0				
20	+1	−1	0	20	0	0	0				
21	+1	−1	+1								
22	+1	0	−1								
23	+1	0	0								
24	+1	0	+1								
25	+1	+1	−1								
26	+1	+1	0								
27	+1	+1	+1								

### 5.2. Central Composite Design (CCD)

The Central Composite design (CCD) includes 2-level FFD (2*^factors^*), a star design (2 × *factors*) and CP, and thus the number of experimental points (*N*) of CCD is calculated using the following Equation (1), where *C*_0_ is the repeated number of the CP. [53].
(1)$$N=2^{factors}+2\times factors+C_{0}$$

CCD evaluates five levels of each factor, which are expressed as −α, −1, 0, +1, and +α, where FFD points are located at level −1 and +1 and star design points are located at level −α and +α, and the CP is located at 0 [53]. The design matrix of CCD for three factors is summarized in Table 7 and Figure 4b. CCD is distinguished into two common types, a face-centered CCD and a circumscribed CCD, according to the value of α [62]. The face-centered CCD is used when |α| = 1 requires three levels per factor, and the circumscribed CCD is used when |α| > 1 requires five levels per factor [62]. To perform the so-called rotatable circumscribed CCD, the equation, |α| = (2*^factors^*)^1⁄4^ is applied to deduce the (−α, +α) of the star design [71]. Therefore, for the number of factors 2, 3, 4, 5, and 6, the |α| values are 1.41, 1.68, 2.00, 2.38, and 2.83, respectively [62]. 

CCD is widely applied to optimize the condition of analysis for plant resources [34,35,36,39]. In the study by Kim et al. [34], two-factors CCD was performed to evaluate the relationship between CAPAs (peak resolution of flavonoids) and CAPPs (column temperature and gradient slope of mobile solvents) using five points (−1.41, −1, 0, +1, +1.41) for each parameter. In another study, Kim et al. [36] performed two set of CCD separately for the simultaneous analysis of paeoniflorin and decursin in herbal medicine by HPLC-PDA, because two target compounds were eluted in prominently different retention times based on their different polarities. Zhang and coworkers applied an AQbD strategy to evaluate the fingerprint of a licorice standard decoction for the purpose of the chemical profiling method [35]. They evaluated the effect of four gradient parameters in the mobile phase solvents to total peak numbers, peak purity, and capacity factor distributions. For the development of a simultaneous HPLC-UV analysis method for the quantitative determination of five saponins in *Panax notoginseng*, Gong et al. [39] utilized a CCD to optimize the gradient elution conditions.

### 5.3. Box-Behnken Design (BBD)

Box-Behnken design (BBD) is a class of second-order rotatable or approximately rotatable designs based on 3-level incomplete factorial design [75,76]. The number of experiments (*N*) is calculated using Equation (2), where *C*_0_ is the repeated number of the CP, that is, fifteen runs are required for three factors with tri-repeated CPs [53].
(2)$$N=\left\{2\times factors\left(factors-1\right)\right\}+C_{0}$$

The BBD can be used as an alternative to CCD or FFD and it is more efficient than the FFD because the experimental points of the BBD designed for three factors are all included in the 3^3^-FFD [75,76]. The BBD is mainly applied when the number of factors is above three and below five [54]. BBD is a combination that does not contain all factors at the highest or lowest levels at the same time. Therefore, these designs help to avoid experiments under extreme conditions where unsatisfactory results may occur. Conversely, they cannot be applied to situations where we want to know the extreme response, i.e., in the vertices of the cube [76]. The matrix of the BBD for three factors is summarized in Table 7 and Figure 4c. For the optimization of parameter conditions for the analysis of medicinal plants, BBD is one of the frequently applied designs [37,41,45,47,49]. Zhang et al. [41] and Tiwari et al. [37] examined the relationship between CAPAs and CAPPs to optimize the conditions of analytical parameters for HPLC separations through BBD experimental data. For the quantification of sugars and monosaccharide derivatives in a mixed plants extract of *Codonopsis pilosula* and *Astragalus membranaceus*, Shao et al. [45] and Wang et al. [47] developed the optimized methods for HPLC-ELSD analysis, respectively. 

### 5.4. Doehlert Design

A Doehlert design fills the space uniformly, and each experimental point has the same distance from the neighboring point [62]. In the Doehlert design, the number of experiments (*N*) is calculated using Equation (3), where *k* is the number of replicated CP [77]. The experiment with two factors required seven experimental points, which are composed of six vertices of a hexagon and the CP (Figure 4d and Table 8), and three factors required thirteen points consisting of a centered dodecahedron with the CP (Figure 4e and Table 8).
(3)$$N=factors^{2}+factors+k$$

**Table 8 plants-11-02960-t008:** The matrix of Doehlert design with two and three factors.

For Two Experimental Factors			For Three Experimental Factors			
No.	X_1_	X_2_	No.	X_1_	X_2_	X_3_
1	0	0	1	1	0	0
2	1	0	2	0.5	0.866	0
3	0.5	0.866	3	0.5	0.289	0.816
4	−1	0	4	−1	0	0
5	−0.5	−0.866	5	−0.5	−0.866	0
6	−0.5	−0.866	6	−0.5	−0.289	−0.816
7	−0.5	0.866	7	0.5	−0.866	0
			8	0.5	−0.289	−0.816
			9	0	0.577	−0.816
			10	−0.5	0.866	0
			11	−0.5	0.289	0.816
			12	0	−0.577	0.816
			13	0	0	0

Unlike the above designs, the level of experimental factors within a Doehlert design varies from factor to factor, i.e., when designing an experiment with two factors, one factor has three levels and the other factor has five levels [53,62]. Deidda et al. [48] used a Doehlert design to evaluate the relationship between CAPPs (column temperature, pH buffer, and flow rate) and CAPAs (total analysis time and critical peak resolution) for the selective determination of cannabidiol and △^9^-tetrahydrocannabinol in original cannabis oil. Ancillotti et al. [42] also used a Doehlert design to develop an LC-MS analysis method for the determination of selected polyphenols contained in *Diospyros kaki*.

## 6. *Specify-Applicable Range*: Method Operable Design Region (MODR)

When the optimization of the analytical parameters is derived, an optimization step of the parameters’ range follows, which uses computer software and virtual screening to determine the method operable design region (MODR, also known as the design space) [21,78,79]. The MODR is the region that satisfies the preset ATP within the allowable variation range of CAPPs, which is established in a multi-dimensional space based on the relationship between CAPPs and CAPAs, so that MODR can provide suitable analytical procedure performance [27,80]. According to the ICH Q14 guideline [18], the range established by the univariate evaluation of a single parameter is defined as a proven acceptable range (PAR). Ideally, MODR (or PAR) combines ATP requirements and the probability that programs meet these criteria with predictive models on the DoE and is validated throughout the procedure lifecycle and refined as needed when new knowledge is gained [79]. The method operable region and continuous improvement process provide robust analytics with regulatory flexibility [14]. Monte-Carlo probability [17,35,38,39,42,44,45,46,47], Bayesian probability [48,49], Desirability function [37,44], Capability analysis [38,46], and Overlay plot [36,40] have been utilized to establish MODR for the analytical procedure of medicinal plants.

## 7. *Planned-Set-of-Controls*: Analytical Procedure Control Strategy

It is important to set appropriate control strategies to ensure analytical procedure performance and quality. The concept of “Control Strategy” emerged in ICH Q8 (R2) guideline [21] and is further developed in ICH guideline Q10 [29] and Q11 [50]. Then, it has been expanded to the analytical procedure field in ICH Q14 guideline ‘Analytical procedure development’ [18]. The analytical procedure control strategy is a planned set of controls that is derived from the properties of the analyte understanding of the MODR. It can be established from the complete statistics collected during the DoE and MODR stages discussed above [27]. Compared to the traditional approach, it may appear that there is no significant difference in the analytical procedure control strategies under the AQbD approach [27]. However, in the traditional approach of developing analytical procedures and control strategies, set points and operating ranges are often strictly set based on observed data to ensure consistent performance of the analytical procedure [50]. For medicinal plants, this limited range setting can easily lead to OOS results because they contain multi-components and can be affected by various factors. A control strategy based on enhanced approaches such as AQbD can provide flexibility in the operating range of analytical parameters to account for the fluctuations [50]. As the analytical procedure control strategy is developed by considering ATP, CAPAs, DoE experimental data, and MODR, it provides a stronger link between the analytical performance and the purpose [27]. The analytical procedure control strategy should be determined prior to the validation guideline [81] and be confirmed after validation is complete [18]. The analytical procedure control strategy includes the analytical procedure parameters to be controlled and the system suitability test (SST) as a part of the analytical procedure description. The analytical procedure description should include the steps required to perform each analytical test, such as the preparation of the samples, reference materials and reagents, the use of apparatus, generation of the formulae and calibration curve for the calculation of the reportable results, and other necessary steps [18].

## 8. *Confirm-the Validity*: Validation of Analytical Procedures

Established Conditions (ECs) are defined in ICH guideline Q12 as legally binding information and are considered necessary to ensure product quality [82], hence ECs for an analytical procedure also should be established, proposed and justified by the applicant and approved by the regulatory authority [18]. ECs can be identified through risk assessment, prior knowledge, and insights from univariate and/or multivariate experiments [18]. Then ECs should be confirmed by validation parameters which are specificity, linearity, accuracy, precision (repeatability, intermediate precision, and reproducibility), range, detection limit (DL), quantitation limit (QL), robustness, and system suitability as per the ICH guideline Q2 (R1) “Validation of Analytical Procedures” [81].

Robustness should be built into the model by including relevant sources of variability related to materials, processes, environment, instrumentation and other factors [18]. For most procedures, robustness assessments are performed during development. If a robustness assessment has already been performed during development, it does not need to be repeated during the validation, as described in ICH Q2 [18]. Model development should minimize prediction errors and provide robust models to consistently ensure long-term performance of multivariate models

## 9. *Prolong-for-Warranty*: Analytical Procedure Lifecycle

Therefore, the lifecycle referred to the overall stages of the products, such as product information, manufacturing processes, sales and marketing [21]. Lifecycle management had not been explicitly addressed in the analytical field, however it has been discussed for a long time, and the concept of lifecycle for analytical procedures was officially been introduced by ICH guideline Q14 ‘Analytical Procedure Development’ [18] and United States Pharmacopeia–National Formulary (USP-NF) ‘Analytical Procedure Life Cycle’ [13] in 2022. Analytical procedure lifecycle can be described as the overall log management of analytical methods established through QRM [11] and validation, such as method design, performance evaluation, and continuous improvement [7]. Through analytical procedure lifecycle management, any anomalies that may arise can be detected and corrected by improving analytical data using the information obtained from the whole process mentioned above [14,30]. Reviewing the results of analytical procedure facilitates the procedure lifecycle management and enables proactive intervention to avoid errors [18]. It also helps to reduce procedures and costs when a change in the approach to validated methods is required and to maintain at the initially set goals by reducing the confusion caused by random variables. 

## 10. *Perspectives*: Challenges and Prospects

The goal of AQbD is to develop a high-quality procedure with robustness that consistently delivers expected performance. The information acquired during risk assessment, method development, optimization, and validation helps to justify the establishment of MODR, which is the design space for the analytical procedure [83].

A variety of analytical methods for medicinal plants have been developed through an AQbD approach, such as the mass spectrometry analysis [84,85], capillary electrophoresis analysis [86,87,88], supercritical fluid chromatography analysis [89,90], gas chromatography analysis [91,92] and so on. Applications of AQbD to the development a HPLC analysis were announced in many studies [36,37,38,39,40,48,49]. According to previous research studies, the general strategies to establish an optimal method of analysis for one or several API(s) [93] were successfully reported. However, very limited examples could be found on the application of the AQbD approach for the optimization of analytical methods for botanical extracts. The botanical extracts have complex and diverse metabolites so the analytical conditions would not be simple. Additionally, the analytical parameters would be difficult to optimize by the DoE technique based on statistic methods. Therefore, diverse approaches to apply AQbD for the development of optimized analytic processes on botanical extract need to be explored in the pharmaceutical industry.

Due to the complexity of natural product ingredients, it was impossible to simultaneously evaluate the effects of various variables in the traditional way to develop the optimized analytical procedure. Through the AQbD approach, the desired ATPs and CAPAs can be set, and the interactions and effects of appropriate variable can be evaluated to determine the conditions in the form of a range, such as MODR and PAR. It can suggest an analytical procedure that is able to more comprehensively evaluate the complexity and diversity of medicinal plants and can be established through the AQbD approach. However, in order to develop medicinal plants as an active pharmaceutical ingredient, the application of DoE to the production and manufacturing process, to secure consistent production of drug substances (or raw materials) with the desired quality, should be conducted. Then, the AQbD-based development of the analytical procedure may easily provide consistent analysis performance.

In the process of developing new drugs, the AQbD approaches have been applied in several stages, such as raw material control [94], bioassay [95,96], stability test [96,97], impurity test [98,99,100], efficacy and safety test [101]. However, there are several limits to apply the AQbD approach in developing new drugs based on medicinal plants, and this is closely related to the limitations of botanical resources as a raw material for drug development [1]. In order to provide the “totality-of-evidence” approach [102] for the quality control of medicinal plants, it is necessary to collect and quantify all possible analytical information from the raw materials to the final drug products. In addition, in the process of obtaining chromatographic information, it is very important to select the appropriate CAPAs and CAPPs to efficiently evaluate a large amount of analysis results because analysis results are not expressed in a single numerical value, but rather are obtained as continuous data or 3D data (e.g., PDA spectra) over time. Therefore, it is very important to ensure that there are no mistakes or omissions at the stage of selecting CAPAs and CAPPs in the experimental design. The fact that the risk assessment and DoE steps in AQbD approach allow us to assess the impact of numerous APPs and their interactions as a whole represents a clear comparative advantage over traditional analysis procedures. However, more consideration and further studies should be done on how to interpret the results of each stage of AQbD and make it more effective for the final decisions.

## Figures and Tables

**Figure 1 plants-11-02960-f001:**
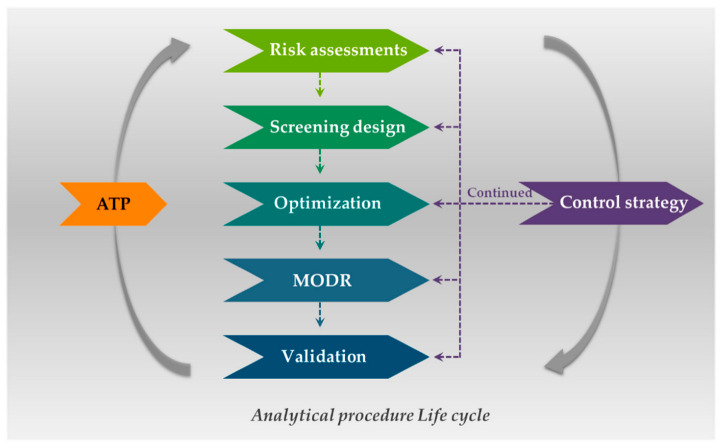
Schematic representation of the steps of the AQbD process: Analytical target profile (ATP), risk assessments, screening design, optimization, method operable design region (MODR), analytical procedure validation, and control strategy.

**Figure 2 plants-11-02960-f002:**
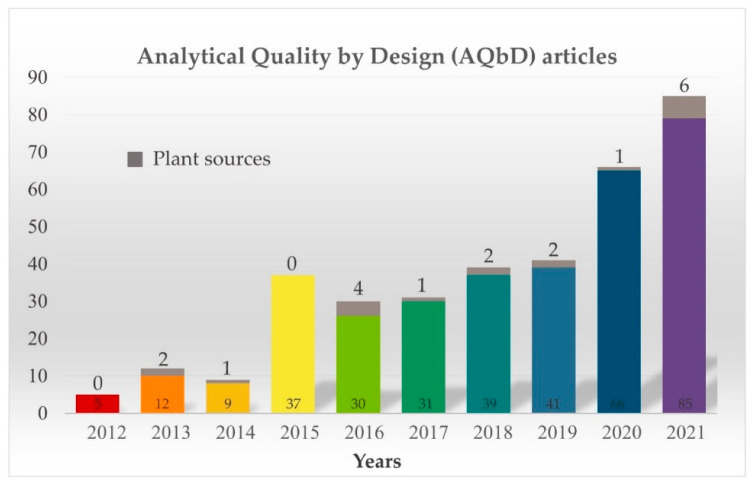
The number of scientific publications by year related to AQbD, searching “Analytical quality by design” (color bars) term in pharmaceutical industry, sorting papers for plant sources (gray bars), Scopus database (2012 to 2021).

**Figure 3 plants-11-02960-f003:**
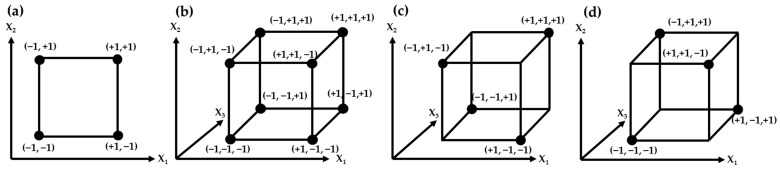
Illustration of two-level FFD matrix for 2-factors (2^2^), (**a**) 3-factors (2^3^), (**b**) 2-level fFD matrix with 3-factors, and (**c**) and its complementary design matrix (**d**).

**Figure 4 plants-11-02960-f004:**
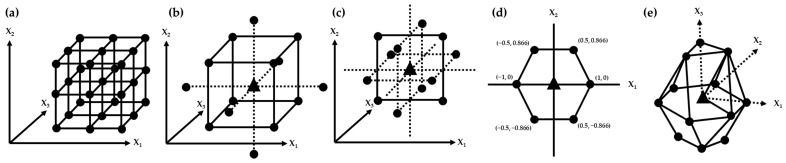
Illustration of 3^3^-Full factorial design (**a**), Central Composite Design with 3-factors (**b**), Box-Behnken design with 3-factors (**c**), and Doehlert design for 2-factors (**d**) and 3-factors (**e**).

**Table 1 plants-11-02960-t001:** The terminology and abbreviations.

Introduction	
QbD	Quality by Design (ICH Q8)
AQbD	Analytical Quality by Design
OOS	Out-of-specification
USFDA	United States Food and Drug Administration
HMP	Herbal Medicine Products
TCM	Traditional Chinese Medicine
ICH	International Council for Harmonization of Technical Requirements for Pharmaceuticals for Human Use
QRM	Quality risk management (ICH Q9)
USP	United States Pharmacopeia
QbT	Quality by Testing
OFAT	One-factor-at-time
PQS	Pharmaceutical Quality System (ICH Q10)
** *Fit-for-Purpose* ** **: Analytical Target Profiles**	
ATP(s)	Analytical Target Profile(s) (ICH Q14)
API(s)	Active Pharmaceutical Ingredient(s)
** *Backbone-of-AQbD* ** **: Risk Assessment**	
APA(s)	Analytical Procedure Attribute(s) (ICH Q14)
CAPA(s) ^a^	Critical Analytical Procedure Attribute(s)
APP(s)	Analytical Procedure Parameter(s) (ICH Q14)
CAPP(s) ^b^	Critical Analytical Procedure Parameter(s)
FMEA	Failure Modes Effects Analysis
RPN	Risk Priority Number
** *Sort-the Main effects-out* ** **: Design of Experiment for Screening factors**	
DoE	Design of Experiment
FFD	Full Factorial Design
fFD	Fractional Factorial Design
PBD	Placket-Burman Design
CP(s)	Center Point(s)
** *Make-the Best conditions* ** **: Design of Experiment for Optimization**	
CCD	Central Composite Design
BBD	Box-Behnken Design
***Specify-Applicable Range*: Method Operable Design Region**	
MODR	Method Operable Design Region (ICH Q14)
PAR	Proven Acceptable Range for Analytical Procedure (ICH Q14) cf. Proven Acceptable Range (ICH Q8)
***Planned-set-of-Controls*: Analytical Procedure Control Strategy**	
SST	System Suitability Test (ICH Q14)
***Confirm-the validity*: Validation of Analytical Procedures**	
EC(s)	Established Condition(s) (ICH Q12)
DL	Detection Limit (ICH Q2)
QL	Quantitation Limit (ICH Q2)
***Prolong-for-Warranty*: Analytical Procedure Lifecycle**	
USP-NP	United States Pharmacopeia–National Formulary

^a^ Harmonized term in this study with a similar meaning as the critical method attribute (CMA) in various studies; ^b^ Harmonized term in this study with a similar meaning as the critical method parameter (CMP) in various studies.

**Table 2 plants-11-02960-t002:** The summary of Appendix A “Applications of AQbD to the development of analytical procedures for plant sources, published from 2016 to 2021”.

Year	Authors	Medicinal Plants	Reference
2021	Zhang et al.	*Ginkgo biloba*	[41]
2021	Tiwari et al.	*Plumbago* species	[37]
2021	Kim et al.	*Daphne genkwa*	[34]
2021	Parab Gaonkar et al.	*Curcuma longa*	[40]
2021	Kim et al.	*Paeonia lactiflora* & *Angelica gigas*	[36]
2021	Deidda et al.	*Cannabis* Species	[43]
2020	Silva et al.	Sugarcane Honey	[44]
2019	Deidda et al.	*Cannabis*-olive oil	[48]
2019	Zhang et al.	*Glycyrrhiza glabra*	[35]
2018	Ancillotti et al.	*Diospyros kaki*	[42]
2018	Shao et al.	*Codonopsis pilosula* & *Astragalus membranaceus*	[45]
2017	Silva et al.	*Saccharum officinarum*	[46]
2016	Sheng-Yun et al.	*Coptis chinensis*	[49]
2016	Wang et al.	*Codonopsis pilosula* & *Astragalus membranaceus*	[47]
2016	Dai et al.	*Panax notoginseng*	[38]
2016	Gong et al.	*Panax notoginseng*	[39]

**Table 3 plants-11-02960-t003:** Characteristics of screening and optimization DoE.

Type of DoE	Experimental Design	General Number of Factors (k)	Levels	Number of Experiments (N)
Screening DoE	Two-level Full factorial design	2 < *k* < 5	2	$2^{k}$
	Two-level fractional factorial design	*k* > 3	2	$2^{k-p}$ *
	Plackett-Burman design	*k* < *N* − 1	2	*N*
Optimization DoE	Three-level full factorial design	2 < *k* < 3	3	$3^{k}$
	Composite central design	2*k* < 5	5	$2^{k}+2k+C$
	Box-Behnken design	3 < *k* < 5	3	2k(k−1)+C
	Doehlert design	2 or 3	Multiple	$k^{2}+k+C$

* *p* is the number of design generators used to separate the design.

**Table 4 plants-11-02960-t004:** Two-level FFD matrix for two (2^2^), three (2^3^), and four (2^4^) experimental factors.

2^2^ Design Matrix			2^3^ Design Matrix				2^4^ Design Matrix				
No.	X_1_	X_2_	No.	X_1_	X_2_	X_3_	No.	X_1_	X_2_	X_3_	X_4_
1	−1	−1	1	−1	−1	−1	1	−1	−1	−1	−1
2	−1	+1	2	−1	−1	+1	2	−1	−1	−1	+1
3	+1	−1	3	−1	+1	−1	3	−1	−1	+1	−1
4	+1	+1	4	−1	+1	+1	4	−1	−1	+1	+1
			5	+1	−1	−1	5	−1	+1	−1	−1
			6	+1	−1	+1	6	−1	+1	−1	+1
			7	+1	+1	−1	7	−1	+1	+1	−1
			8	+1	+1	+1	8	−1	+1	+1	+1
							9	+1	−1	−1	−1
							10	+1	−1	−1	+1
							11	+1	−1	+1	−1
							12	+1	−1	+1	+1
							13	+1	+1	−1	−1
							14	+1	+1	−1	+1
							15	+1	+1	+1	−1
							16	+1	+1	+1	+1

**Table 5 plants-11-02960-t005:** Two-level fFD matrix for three (2^3−1^), four (2^4−1^), and five (2^5−2^) experimental factors.

2^3−1^ Design Matrix *^A^				2^4−1^ Design Matrix *^B^					2^5−2^ Design Matrix *^C^					
No.	X_1_	X_2_	X_3_	No.	X_1_	X_2_	X_3_	X_4_	No.	X_1_	X_2_	X_3_	X_4_	X_5_
1	−1	−1	+1	1	−1	−1	−1	−1	1	−1	−1	−1	+1	+1
2	−1	+1	−1	2	+1	−1	−1	+1	2	+1	−1	−1	−1	−1
3	+1	−1	−1	3	−1	+1	−1	+1	3	−1	+1	−1	−1	+1
4	+1	+1	+1	4	+1	+1	−1	−1	4	+1	+1	−1	+1	−1
				5	−1	−1	+1	+1	5	−1	−1	+1	+1	−1
				6	+1	−1	+1	−1	6	+1	−1	+1	−1	+1
				7	−1	+1	+1	−1	7	−1	+1	+1	−1	−1
				8	+1	+1	+1	+1	8	+1	+1	+1	+1	+1

*^A^ X_3_ = X_1_ × X_2_; *^B^ X_4_ = X_1_ × X_2_ × X_3_; *^C^ X_4_ = X_1_
× X_2_; X_5_ = X_1_ × X_3_.

## Data Availability

Not applicable.

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
