# Peer review of "Analytical Quality by Design (AQbD) Approach to the Development of Analytical Procedures for Medicinal Plants"

_plants, 2022, doi:10.3390/plants11212960_

Round 1

Reviewer 1 Report

The review manuscript is generally written in a clear and concise way, and also illustrations are well done. It would be of broad interest to “plants” readers and the medicinal plants industry.  And the figures in this manuscript are easy to understand.

I only have a few suggestions:

1. There are too many sub-title, it is possible to combine some of them into one. For example, 6. Specify-Applicable Range: Method Operable Design Region (MODR) and 7. Planned-set-of-Controls: Analytical Procedure Control Strategy

2. In figure 2, it is better also to show the No. of Analytical quality by design” (color bars) term in the pharmaceutical industry in the “belowed color region”

3. As the abbreviation was used for the first time, the authors should explain it. Such as on page 6 line 186  “HPLC” and in line 189 “ELSD”. Please check all of the abbreviations in the manuscript. 

Author Response

There are too many sub-title, it is possible to combine some of them into one. For example, 6. Specify-Applicable Range: Method Operable Design Region (MODR) and 7. Planned-set-of-Controls: Analytical Procedure Control Strategy

A: Thanks for your kind suggestion. We organized our manuscript by dividing the important steps of AQbD into subheadings. The establishment of control strategy is considered as an important step to be determined before entering the validation step after obtaining results through experimental steps such as DOE and MODR, so it would be better to classify it independently.

  1. In figure 2, it is better also to show the No. of Analytical quality by design” (color bars) term in the pharmaceutical industry in the “belowed color region”

A: We marked the number of publications in the pharmaceutical industry on the graph.

  1. As the abbreviation was used for the first time, the authors should explain it. Such as on page 6 line 186  “HPLC” and in line 189 “ELSD”. Please check all of the abbreviations in the manuscript. 

A: We provided the full names of the abbreviations in the manuscript.

Reviewer 2 Report

Line Number

Comment Type

Comment

N/A

General

Analytical method and procedure are used interchangeably throughout the paper – propose the authors use the ICHQ2/Q14 term (analytical procedure) throughout unless intending to differentiate between the method and procedure.

16, 17

English/Grammar correction

Change sentence to: :

Scientific regulatory systems with the suitable analytical methods for monitoring quality, safety, and efficacy is essential in medicinal plant drug discovery.

17

English/Grammar correction

There have been a few attempts of adopting of analytical quality by design (AQbD) in medicinal plants analysis over the last few years strategy. AQbD  which is a holistic method development approach for understanding an analytical procedure from risk assessment to lifecycle management were proposed

22-23

English/Grammar correction

Change “it is difficult to set the diverse steps…” to “it is difficult to follow all the AQbD workflow steps…”

26-29

English/Grammar correction

In this review, various applications of AQbD to medicinal plant analytical procedures are discussed

Rest of the sentence needs to be written more clearly to line 29

73

Incorrect reference

USP<1220> has been finalised and published – reference should be updated to latest version of Pharmacopeial Forum

81-82

Text is incorrect

Change “which is supposed to be announced in the ongoing public consulting ICH new guideline Q14 on analytical procedure development [18].” to “which is described in the Step 2 draft of ICHQ14 on analytical procedure development [18].

Recommend that authors highlight that ICHQ14 calls AQbD the ‘enhanced approach’

83

(also 762)

Incorrect reference

I think the authors mean to reference to the latest draft version of ICHQ14 here (ref 18): ICH_Q14_Document_Step2_Guideline_2022_0324.pdf

Similar issue with ref 11, 18, 29, 50, 82

84

English/Grammar correction

Change ‘USFDA’ to ‘FDA’ – as it is known that FDA is the US health authority

97-101

Text is incorrect

The ATP is linked to a measurement of a quality attribute (see Q14 definition). It is agnostic of technology and method specifics. Authors need to update this sentence.

Suggest authors reference following paper which explain link between technology agnostic ATP and selection of analytical technology:

Selection of Analytical Technology and Development of Analytical Procedures Using the Analytical Target Profile | Analytical Chemistry (acs.org)

102-103

Clarify text

CAPAs and CAPPs are related to specific analytical procedures and not the ATP. Text needs clarifying

109

English/Grammar correction

Change “The great risk…” to “Parameters assessed to have the highest risk are selected…”

112

Text is incorrect

Change “is utilised” to “can be utilised” as DoE studies are not always required

115

English/Grammar correction

Change critic to critical?

116

English/Grammar correction

Remove synergetic

I stopped reviewing the manuscript after line 116 as extensive editing of the English used (especially grammar) is required. I suggest the authors enlist the help of a ntive speaker to perform a thorough review of how the paper is written.

The paper attempts to sumamrise what the AQbD approach entails and how it has been applied to analytical procedures supporting medicinal plant drug discovery/development. There is nothing novel in sumamrising the AQbD approach. Readers could just look at the latest draft version of ICHQ14 for example.  

Author Response

Reviewer 2

Line Number

Comment Type

Comment

Answer

N/A

General

Analytical method and procedure are used interchangeably throughout the paper – propose the authors use the ICHQ2/Q14 term (analytical procedure) throughout unless intending to differentiate between the method and procedure.

Except for the case where 'method' is specifically used in the text, it has been unified to 'procedure'.

16, 17

English/ Grammar correction

Change sentence to: :

Scientific regulatory systems with the suitable analytical methods for monitoring quality, safety, and efficacy is essential in medicinal plant drug discovery.

It was revised according to your recommendation.

17

English/ Grammar correction

There have been a few attempts of adopting of analytical quality by design (AQbD) in medicinal plants analysis over the last few years strategy. AQbD  which is a holistic method development approach for understanding an analytical procedure from risk assessment to lifecycle management were proposed

It was revised according to your recommendation.

22-23

English/ Grammar correction

Change “it is difficult to set the diverse steps…” to “it is difficult to follow all the AQbD workflow steps…”

It was revised according to your recommendation.

26-29

English/ Grammar correction

In this review, various applications of AQbD to medicinal plant analytical procedures are discussed

Rest of the sentence needs to be written more clearly to line 29

It was modified to give clearer meaning.

73

Incorrect reference

USP<1220> has been finalised and published – reference should be updated to latest version of Pharmacopeial Forum

The reference has been updated to the latest version.

81-82

Text is incorrect

Change “which is supposed to be announced in the ongoing public consulting ICH new guideline Q14 on analytical procedure development [18].” to “which is described in the Step 2 draft of ICHQ14 on analytical procedure development [18].

Recommend that authors highlight that ICHQ14 calls AQbD the ‘enhanced approach’

It was revised according to your recommendation

83

(also 762)

Incorrect reference

I think the authors mean to reference to the latest draft version of ICHQ14 here (ref 18): ICH_Q14_Document_Step2_Guideline_2022_0324.pdf

Similar issue with ref 11, 18, 29, 50, 82

All the reference has been updated to the latest version with the web addresses where the pdf documents can be found.

84

English/Grammar correction

Change ‘USFDA’ to ‘FDA’ – as it is known that FDA is the US health authority

It was revised according to your recommendation

97-101

Text is incorrect

The ATP is linked to a measurement of a quality attribute (see Q14 definition). It is agnostic of technology and method specifics. Authors need to update this sentence.

Suggest authors reference following paper which explain link between technology agnostic ATP and selection of analytical technology:

Selection of Analytical Technology and Development of Analytical Procedures Using the Analytical Target Profile | Analytical Chemistry (acs.org)

The sentence was revised to deliver that the characteristics of ATP. Suggested reference was added for this revision.

102-103

Clarify text

CAPAs and CAPPs are related to specific analytical procedures and not the ATP. Text needs clarifying

The text was revised according to your suggestion.

109

English/Grammar correction

Change “The great risk…” to “Parameters assessed to have the highest risk are selected…”

It was revised according to your recommendation

112

Text is incorrect

Change “is utilised” to “can be utilised” as DoE studies are not always required

It was revised according to your recommendation

115

English/Grammar correction

Change critic to critical?

It was revised according to your recommendation

116

English/Grammar correction

Remove synergetic

It was revised according to your recommendation

I stopped reviewing the manuscript after line 116 as extensive editing of the English used (especially grammar) is required. I suggest the authors enlist the help of a ntive speaker to perform a thorough review of how the paper is written.

A: We revised substantially the entire manuscript with the help of a native speaker.

The paper attempts to sumamrise what the AQbD approach entails and how it has been applied to analytical procedures supporting medicinal plant drug discovery/development. There is nothing novel in sumamrising the AQbD approach. Readers could just look at the latest draft version of ICHQ14 for example.  

A: Please understand that the purpose of this paper is not to provide new knowledge about AQbD, but to introduce why this well-established technology is necessary for natural products research and what to pay attention to in order to apply it. To this end, all previously published AQbD studies related to medicinal plants were organized and shown in actual stages of AQbD to provide ease of application for researchers who are not familiar with this technology.

Round 2

Reviewer 2 Report

Thank you for reviewing the English grammar in the paper. It reads much better.

See attached for minor  corrections which are still required

I have also recommended further references which will further improve the publication 

Author Response

Thank you very much for your kind and thoughtful correction. I revised it to reflect all the points you suggested.